# Impact of Satellite and In Situ Data Assimilation on Hydrological Predictions

**Jude Lubega Musuuza** [1,*]**, David Gustafsson** [1]**, Rafael  Pimentel** [2] **and Louise Crochemore** [1] **and Ilias Pechlivanidis** [1]

[1]    Swedish Meteorological and Hydrological Institute, 60176 Norrköping, Sweden;
      David.Gustafsson@smhi.se (D.G.); louise.crochemore@smhi.se (L.C.); ilias.pechlivanidis@smhi.se (I.P.)

[2]    Fluvial Dynamics and Hydrology Research Group, Andalusian Institute for Earth System Research,
      University of Córdoba, 14071 Córdoba, Spain; z02piler@uco.es

[*]    Correspondence: jude.musuuza@smhi.se; Tel.: +46-11-4958000

**Abstract:** The assimilation of different satellite and in situ products generally improves the hydrological model predictive skill. Most studies have focused on assimilating a single product at a time with the ensemble size subjectively chosen by the modeller. In this study, we used the European-scale Hydrological Predictions for the Environment hydrological model in the Umeälven catchment in northern Sweden with the stream discharge and local reservoir inflow as target variables to objectively choose an ensemble size that optimised model performance when the ensemble Kalman filter method is used. We further assessed the effect of assimilating different satellite products; namely, snow water equivalent, fractional snow cover, and actual and potential evapotranspiration, as well as in situ measurements of river discharge and local reservoir inflows. We finally investigated the combinations of those products that improved model predictions of the target variables and how the model performance varied through the year for those combinations. We found that an ensemble size of 50 was sufficient for all products except the reservoir inflow, which required 100 members and that in situ products outperform satellite products when assimilated. In particular, potential evapotranspiration alone or as combinations with other products did not generally improve predictions of our target variables. However, assimilating combinations of the snow products, discharge and local reservoir without evapotranspiration products improved the model performance.

**Keywords:** data assimilation; ensemble Kalman filter; satellite data; in situ data; hydrological predictions

## 1. Introduction

Climate change, air quality and environmental degradation are currently among the most pressing challenges [1]. Monitoring the status for action requires observations of the atmosphere, land, oceans and the cryosphere. Mountainous regions are the most sensitive to climate change but are poorly observed due to complex terrain and accessibility challenges [2–4]. Ground-based (in situ) observations are usually coarse and prohibitively expensive to maintain at a high quality. In fact, the number of stations is declining globally with many being taken out of service [5–7]. Remote sensing provides the most efficient alternative observations at medium to large scales [8]; from continental [9] to global [10].

Satellite data can provide an added value in hydrological modelling and services by, for example, improving accuracy and spatial detail in hydrological model estimation or forecasted variables provided by climate services [3,11–13]. The set-up of the hydrological models used in most cases is based on dynamic forcing, land cover classifications and parametrizations of vegetation dynamics that are partially (or even entirely) derived from some remote sensing. Satellite data are used to

varying degrees in model calibration [14–16], validation [9,10,17] and data assimilation [3,4,12,13,18]. Karthikeyan et al. [19] outline the achievements during the evolution of satellite product acquisition and retrieval that have ultimately led to the availability of complex data-sets. The utility of these data in hydrological modelling can vary as a function of dominant hydrological processes, while opportunities for improvement of hydrological models and water services include the use of a wider range of satellite products in model setup, parameter estimation, model evaluation and data assimilation. For instance, in data sparse regions, satellites can help improve the quality of forcing inputs [11,20,21], while vegetation and snow/ice products can be used to initialize the hydrological models and classify the landscape into land use classes.

Satellite products are valuable in water resources modelling and monitoring for: (1) dynamic forcing, (2) a priori parameter estimation, (3) model set-up/development, (4) model evaluation, and (5) data assimilation, including both non-sequential (such as parameter calibration) and sequential techniques (i.e., state updating). Satellites provide spatially continuous fields of water quality and hydrological parameters e.g., turbidity, algal blooms, soil and vegetation. The use of satellite products in water resources management services provides a useful feedback that can be used to evaluate the performance of hydrological models [3,13,18]. The feedback also helps to: (i) understand how and against which observations model estimates are most usefully compared, (ii) point towards model structural inconsistencies (systematic differences can subsequently lead to improvements in model structure and/or parametrization), (iii) add value to the development of data assimilation techniques, and (iv) produce model results that are "right for the right reasons".

Data assimilation is a discipline that seeks to optimally combine numerical models with observations. There are various applications in which data assimilation aims to e.g., (i) interpolate sparse observations using knowledge of the system being investigated; (ii) estimate the optimal size of a system; (iii) determine the initial conditions for a forecast model; and (iv) train model parameters based on observed data. Here, we used data assimilation to quantify the deviation of numerical model outputs from the true states. Data assimilation was originally developed and used within the atmospheric sciences [22] and oceanography [23] communities but is now used in other branches of science including hydrology.

Data assimilation is accomplished by variational e.g., 3D-Var, 4D-Var [24–26] and sequential methods e.g., Kalman filters (KF) [27–29]. Sequential methods utilise the least-squares approach to choose the final estimate such that the uncertainty of the final state is minimised [30]. The classic KF method is limited to linear models but that can be overcome by the extended (EKF) and ensemble (EnKF) Kalman filters [12,31–33] among others. The KF approach requires the multiplication and inversion of potentially large matrices, which can very easily become a prohibitive computational bottleneck. The method assumes that the observed and modelled errors are independent and follow Gaussian distributions, which is only true for physically unbounded variables like temperature. The distributions require analytical transformations for semi-bounded e.g., discharge and bounded variables e.g., fractional snow cover (FSC) [4,34]. The simpler EnKF yields competitive results to the more complex variational methods like the 4D-Var [35]. Please see section A.1 for a detailed mathematical treatment of the ensemble Kalman filter.

Soil moisture (SM) is the most widely studied variable in hydrological data assimilation, possibly due to its strong control on the partitioning of moisture and energy fluxes that regulate weather and climate [36]. Other commonly assimilated hydrological observations include stream discharge [37], groundwater elevations [38], evapotranspiration [39], fractional snow cover [40] and snow water equivalent [41].

In this work, we applied the EnKF method to assimilate the MODIS accumulated 8-day actual and potential evapotranspiration (AET and PET, respectively); and the daily CRYOLAND optical satellite FSC and passive microwave snow water equivalent (SWE); and river discharge and local inflows into hydropower reservoirs with the aim of improving the estimates for the latter two variables. Most previous studies e.g., Li et al. [37], Li et al. [38], He et al. [39], Pimentel et al. [40], De Lannoy et al. [41]

assimilated one variable. Few studies assimilated two variables, e.g., Liu et al. [42] assimilated satellite FSC and snow depth; Rasmussen et al. [43] assimilated groundwater heads and stream discharge; while Ines et al. [44] assimilated satellite SM and Leaf Area Index. Pressure head, soil moisture and stream flow data were assimilated on a synthetic catchment in Camporese et al. [45] and in Camporese et al. [46] to test the numerical performance of their schemes. The assimilation of multiple data sets to achieve a practical goal has been missing until now.

To the best of our knowledge, this work is the first to conduct data assimilation of multiple data products at different spatial and temporal scales. Discharge and reservoir inflow, the target variables in this study are very important for the snow-melt driven hydropower industry in northern Sweden. We aim to answer the following questions:

1. What ensemble size of the EnKF assimilation scheme optimises the performance of our type of model?
2. Which product combinations improve predictions of both discharge and reservoir inflow?
3. How does the model performance change over the year for different product combinations?

Section 2 describes the study area and the data used in this study. Section 3 presents the methods and the hydrological model E-HYPE. The results are presented in Section 4, followed by a discussion in Section 5 and conclusions in Section 6.

## 2. Study Area and Data

### 2.1. Study Area

We conducted the study in the Umeälven River catchment located in northern Sweden (Figure 1). It has an area of 26000 km$^2$ with a mean elevation of 510 m above mean seal level and a runoff regime dominated by snow melt during the spring and summer. Annual precipitation ranges from 542 to 874 mm with a mean value of 691 mm during the period 2000–2015. For the same period, daily mean temperature oscillated between −2.1 to 2.3 °C with a mean value of 0.01 °C. The dominant land use types are coniferous forest and open land with vegetation, covering 38 and 32% of the catchment area, respectively. Moraine is the dominant soil constituting almost 60% of the catchment area. The area is extensively used for hydropower production with 19 hydro-power plants whose temporal needs are almost the reverse of the flow regime. Consequently, the storage of water from the snow melt season to the next winter in large reservoirs high up in the head-waters is the most important constraint to take into account for reservoir management.

### 2.2. Data

There are 14 gauging stations with long daily flow records in the area: five on the unregulated Vindelälven arm (asterisks in Figure 1) and nine on the main regulated Umeälven branch (crosses in the figure). However, only the river discharge data at the unregulated stations, available from the SMHI observation network was used to calibrate and assess the performance of the model. There are also nine forecast regions with daily local reservoir inflow time series covering the period 1981–2015 available from the independent regulator company Vattenregleringsföretaget (VRF) AB. Historical time-series of daily precipitation and air temperature values from the HydroGFD dataset [47] for the period 1981–2015 were used to force the model.

Satellite-based snow data representing fractional snow cover (FSC) and snow water equivalent (SWE) were obtained from a service provided by the EU FP7 CryoLand project [48]. The FSC data is based on optical sensors on the NASA TERRA Y AQUA satellite at 8-day temporal resolution on a 500 m grid [49]. The quality of the data is limited by the presence of clouds and low solar angles, which often occurr in northern Scandinavia [50]. We rejected FSC data if the fraction of cloud free (or other temporary errors) cells was lower than 25%. The SWE is based on passive microwave data from several missions at a product resolution of 25 km [51]. The data is independent of cloud and illumination but there are known sensor saturation problems in complex terrains and sensitivity

to wet snow conditions [4]. We consequently rejected data in the headwater mountains which are characterized by complex terrains and in the coastal areas of the domain where wet snow conditions often prevail (hatched regions in Figure 1). We also used the satellite-based 8-day aggregated MODIS actual and potential evapotranspiration [10]. The average value was applied to all days within an aggregation period.

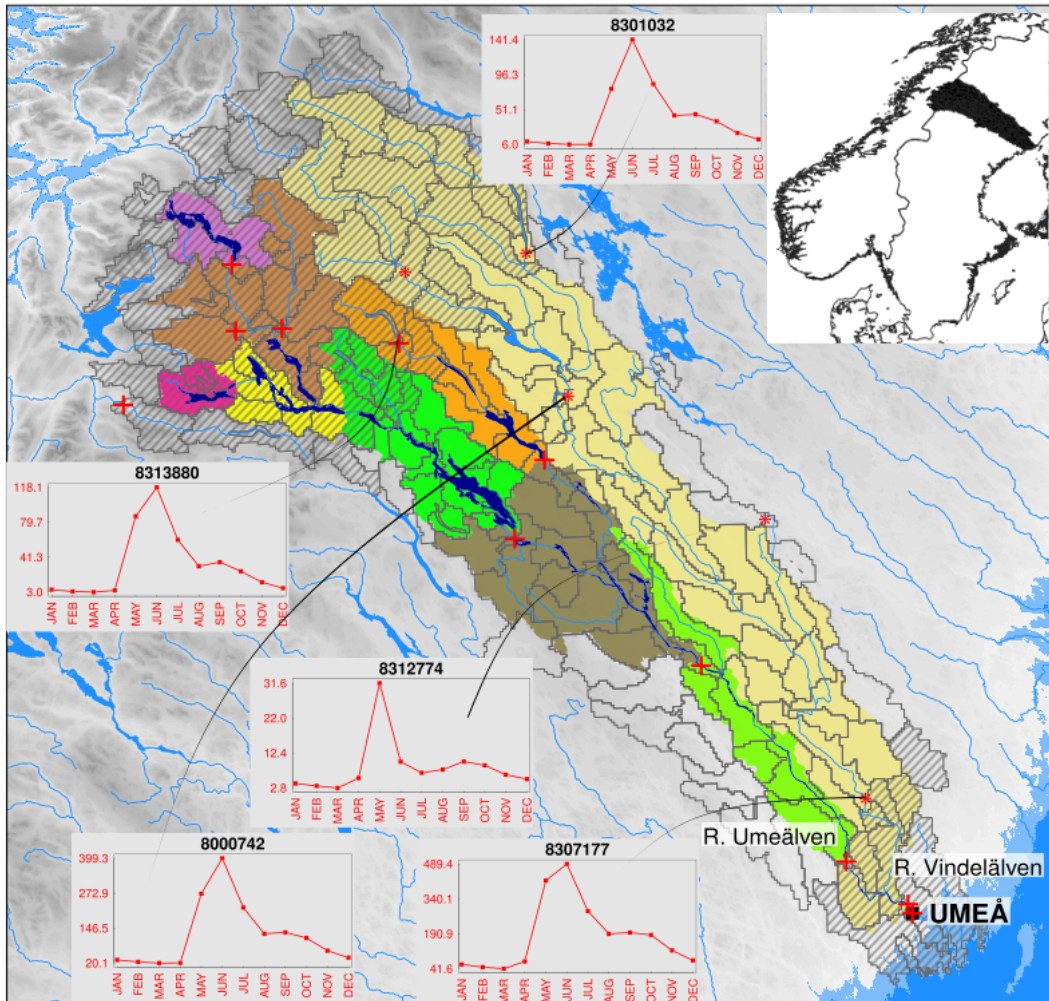

**Figure 1.** The study area with the regulated main Umeälven tributary and the unregulated upper Vindelälven arm coloured light yellow. The red asterisks and crosses are the unregulated and regulated gauging stations, respectively and the coloured polygons are the 9 local inflow regions. The hatched areas in the head water and coastal regions are where the satellite passive microwave snow product is known to be erroneous. The overlaid grey-bounded polygons are the sub-basins in the catchment. The curves are the annual regimes at the five unregulated stations, the numeric headings are the station IDs used in Figure 7.

## 3. Methods

### 3.1. Hydrological Model

A sub-model covering the Umeälven river basin extracted from the pan-European hydrological E-HYPE model [52] was the basis for the case study analysis. Spatial overlays of the sub-basin polygons of the E-HYPE model and the reservoir inflow forecast areas (as shown in Figure 1) were used to derive simulated reservoir local inflows from the model corresponding to the available observations provided by the regulator VRF. Local inflow is the sum of snowmelt and runoff from the local catchment area

between the outlet of an upstream reservoir (if existing) and the outlet of the most downstream reservoir in the forecast area and the net precipitation on the reservoirs within it. It is computed as the residual between the reservoir outflow and inflow; and the change in reservoir storage volume.

The parameters in the model E-HYPE are either defined for the entire domain or are dependent on land use or soils. A subset of the influential parameters to snow processes is shown in Table 1 along with their brief descriptions and ranges.

**Table 1.** A selection of E-HYPE model snow parameters and their ranges.

| Parameter | Description | Range |
|-----------|-------------|-------|
| fscmax | Maximum snow covered area fraction [-] | [0.5, 0.95] |
| cmrefr | Refreeze efficiency compared to the degree-day snow melt factor [-] | $[10^{-3}, 0.1]$ |
| fsceff | Efficiency of snow cover to influence snow melt and evaporation [-] | [0, 0.8] |
| fscdist0 | Minimum snow distribution factor [-] | [0, 0.8] |
| fscdist1 | Std coefficient for snow distribution factor [m$^{-1}$] | [0, 0.3] |
| fepotsnow | Fraction of snow-free potential evapotranspiration [-] | [0, 0.6] |
| cmlt | Degree-day factor [mm·day$^{-1}$·°C$^{-1}$] | [0, 6] |
| ttmp | Threshold temperature for snow melt, [°C] | [−0.5, 3] |
| cmrad | Coefficient for radiation snow melt [mm· m$^{-2}$· MJ$^{-1}$] | [0,0.8] |

### 3.2. Model Performance

The E-HYPE model was previously calibrated on river discharge data [52]. Here, we introduced the four satellite datasets and re-calibrated the parameters in Table 1 for the period 2012–2013. We sampled the parameters over the indicated ranges and optimised them using the Differential Evolution Markov Chain algorithm [53] with 25 populations and 100 generations. All available observation data i.e., discharge ($Q_{ob}$), reservoir inflow (INFLOW), fractional snow cover (FSC), snow water equivalent (SWE), actual ET (AET) and potential ET (PET) were used in the parameter optimisation.

We chose the Kling-Gupta efficiency (KGE) [54] as the performance metric because it enables us to access different aspects (timing, bias and variability) of the modelled time series. It is defined as

$$KGE = 1 - \sqrt{(\rho - 1)^2 + (\alpha - 1)^2 + (\beta - 1)^2}, \tag{1}$$

where $\rho$ is the Pearson correlation coefficient, $\alpha = \dfrac{\sigma_m}{\sigma_o}$ is the variability in the modelled and observed quantities, where $\sigma$ is the standard deviation and subscripts $m$ and $o$ denote the modelled and observed quantities, respectively. The ratio $\beta = \dfrac{\mu_m}{\mu_o}$ is the bias term where $\mu$ is the mean. The objective is to maximise *KGE* on $[-\infty, +1]$.

### 3.3. Data Assimilation with the Ensemble Kalman Filter

We used the ensemble Kalman filter for the data assimilation experiment. It is a sequential algorithm that uses a finite number of members to represent the error statistics. The error covariance matrices are computed from an ensemble and propagated forward by the dynamic model. This set-up requires an infinite number of members in the ensemble, a linear dynamic model and Gaussian errors. These requirements are usually violated since the use of very many ensemble members quickly becomes a computational bottleneck in large domains or with long simulation runs. The normally-distributed error assumption is also invalid for physically bounded variables [4]. Here, we used the *lognormal* and *logit-normal* transformations [34] for semi-bounded and bounded variables, respectively.

At each time step in the assimilation run, we introduced an ensemble of random but finite perturbations in the forcing (precipitation and temperature) and observations (local reservoir inflows, river discharge, and the satellite products) that result into randomly generated model trajectories.

The perturbations in the forcing are spatially correlated fields that help to propagate information to locations where it may be missing [55]. The spatial correlation is controlled by the so-called spatial localization, which suppresses superfluous covariances that can arise in mountainous regions (please see Magnusson et al. [12]).

In the next step, the ensemble of model states and fluxes was propagated through the dynamic model and transformed to the observation space, where the predicted observation and observation error terms were obtained. The localization was included by modifying the error covariance matrix terms present in the Kalman gain matrix (see Equation (A5) in Appendix A.1). These were multiplied by two scalar matrices that restrict the influence of the observations on updated variables using element-wise products. One ensures that the distance-dependent functions used to define the scalar matrices diminish to zero at appropriate separation lengths. The complete list of parameters that control the ensemble Kalman filter in data assimilation is presented in Table A2.

The errors in the gain matrix provide an indication on the quality of the assimilation. To access this, we defined a gain $\Omega$ in the KGE resulting from an assimilation:

$$\Omega = \frac{\omega^\star - \omega}{1 - \omega} \, , \tag{2}$$

where $\omega^\star$ and $\omega$ are the performances with and without data assimilation, respectively. The improvements in the individual components of the KGE could also be viewed as performance gains. The gains are considered relative to the open-loop simulation in which no data is assimilated.

From $Q_{ob}$, INFLOW, FSC, SWE, AET and PET, we obtained 57 additional unique combinations, which when treated as autonomous variables resulted in 63 assimilations shown in Table A1. We did this to identify the datasets that improved model predictions of stream discharge and reservoir inflow when assimilated alone or as combinations.

In addition to the enKF ensemble size, the quality of the data assimilation depends on the minimum and maximum of the perturbations, the localisation length and the minimum allowable standard deviation in the bounded variables. However, we performed a sensitivity investigation for the ensemble size only. That way, we avoided the usual subjectivity surrounding the selection of the ensemble size for the Kalman filters.

## 4. Results

### 4.1. Benchmarking Model Performance

We ran the model with the optimised parameters without assimilation to obtain the starting point for the open-loop simulation, against which assimilations were compared. The spatial model performance with the optimised parameters for all datasets is shown in Figure 2.

The model adequately estimated the FSC and $Q_{ob}$ in all parts of the study area. The variability and correlation in AET were also well reproduced although the variable itself was consistently underestimated. The variability and correlation in INFLOW were also well reproduced in all but one inflow region.

The model underestimated the average SWE in the coastal and highland areas and the variability in the highlands. This suggests that the passive microwave SWE data errors related to complex terrain and wet coastal conditions could not be corrected by perturbing the parameters. Consequently, the highland and coastal values were removed from the assimilated SWE data but a large spatial localization (correlation length scale) was applied in order to propagate the lowland SWE information to the highland and coastal areas. The model also underestimated the PET from the mid to high elevation areas of the domain, hinting at the possibility of very high MODIS PET values. Ruhoff et al. [10] pointed out that same problem during the validation of the MODIS ET products against eddy flux tower records. The bias in that dataset would need to be investigated and quantified but such investigations were beyond the scope of this work.

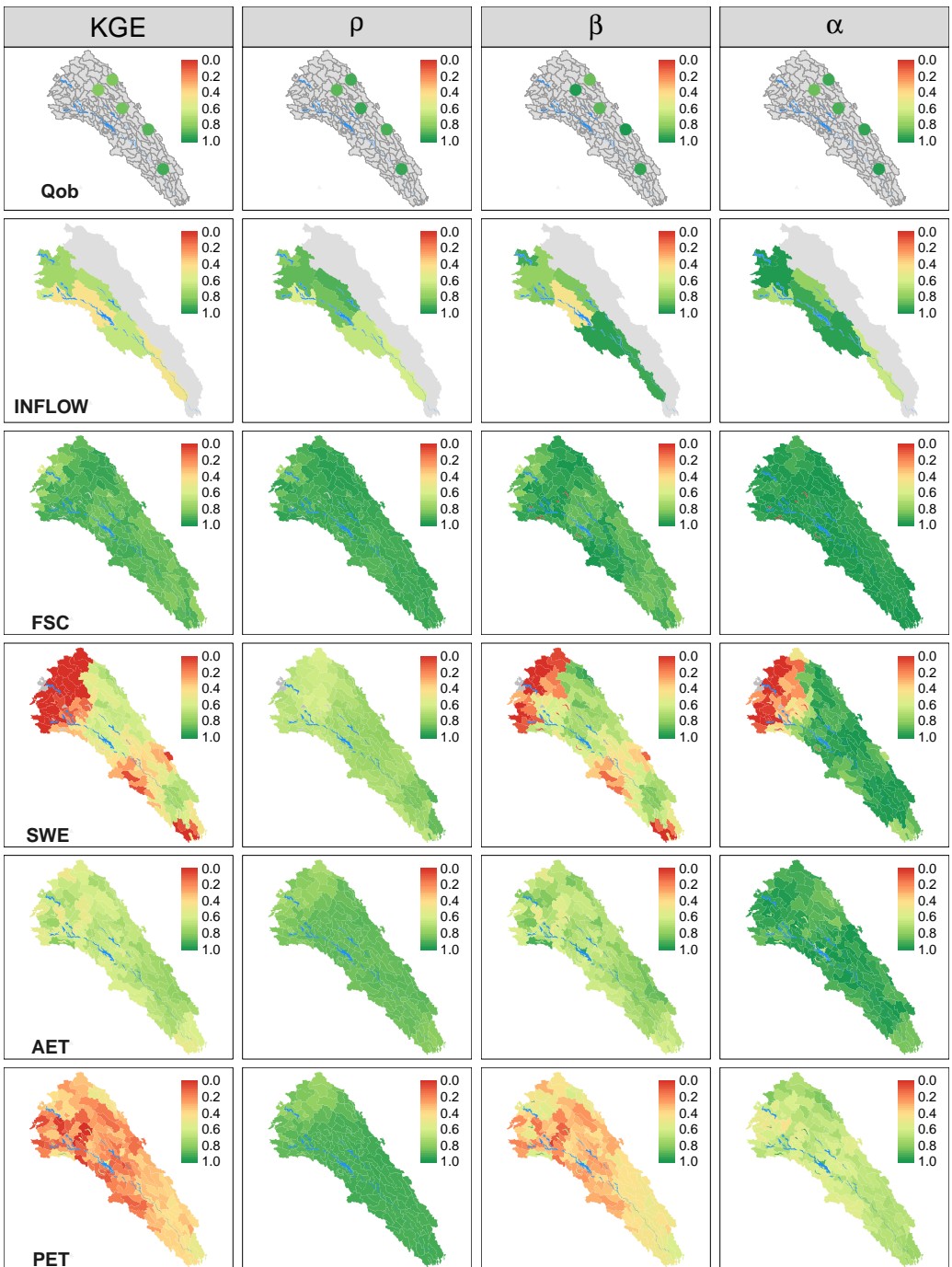

**Figure 2.** The model performance with the optimised parameter set for the river discharge ($Q_{ob}$), local reservoir inflows (INFLOW), FSC, SWE, AET and PET. The columns are the KGE, Pearson correlation coefficient ($\rho$), and the bias ($\beta$) and variability ($\alpha$) coefficients, respectively. Only the regulated main arm is shown in the INFLOW row, the grey area is the unregulated Vindelälven area.

*4.2. Data Assimilation: Sensitivity to Ensemble Size*

We also assessed the sensitivity of the model performance to the ensemble size in data assimilation with the ensemble Kalman filter. We assimilated the same number of data points (namely the unregulated gauge data, the forecast area inflows and the sub-basin FSC, SWE, AET and PET) and only varied the ensemble sizes to 25, 50, 100, 250, 500 and 1000 members. Overall, the optimum ensemble size depended on the variable being assimilated (Figure 3). The results showed that the KGE and its components for FSC, SWE, AET and PET assimilations required between 20 and 25 members to stabilise

above the open-loop values. The same minimum number of members was needed for the three KGE coefficients when $Q_{ob}$ was assimilated. However, the alpha coefficient required 100 members when Qob was assimilated to attain the maximum value that was below the 0.9 for the open-loop. When the INFLOW alone drove the assimilation, KGE, $\rho$ and $\beta$ achieved their maxima with 100 members although the latter was always below the open-loop, which was already very high at around 0.9. The $\alpha$ coefficient deteriorated with increasing ensemble size but it was equal to the open-loop with 50 to 100 members. Based on the above results we set the ensemble size to 100 members for the remaining investigations. In any case, it was hoped that the dataset combinations would require less than 100 members.

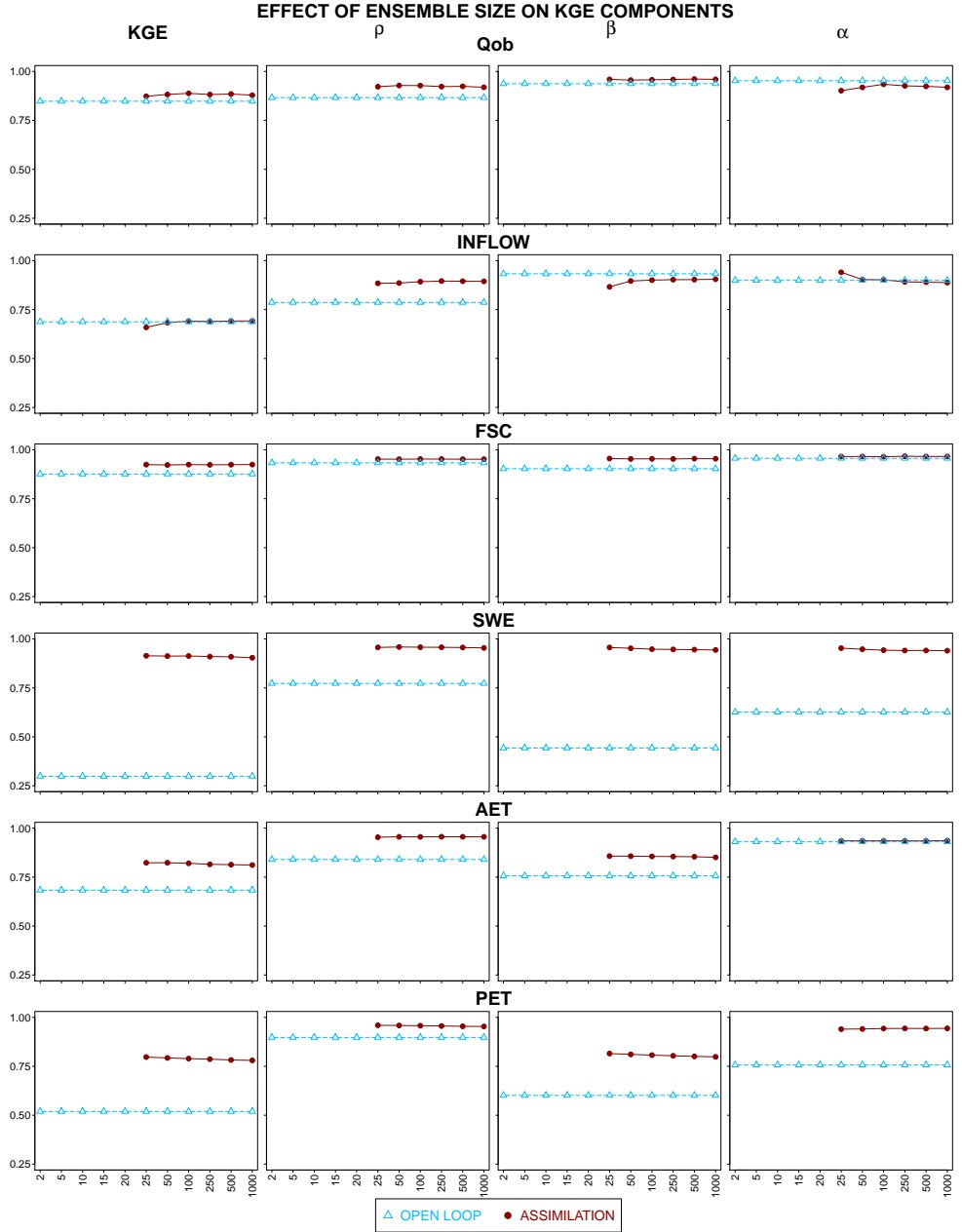

**Figure 3.** The effect of varying the ensemble size on the KGE and its components (columns) for the six assimilated products (rows). The filled red markers are the assimilations and the horizontal blue dashed lines with hollow triangles are the corresponding open-loop simulations. The reservoir inflow and FSC assimilations crashed with less than 25 members in the ensemble.

### 4.3. Data Assimilation: Performance from Single Variables

We next investigated the gain in model performance by assimilating the six products individually. Figure 4 shows that the median gain in KGE and improvements in $\rho$ and $\alpha$ were above zero for all products. The improvement in beta with INFLOW assimilated was very low, possibly because of the already very high value of the open-loop simulation (25th percentiles of all coefficients were below the open-loop simulation). The assimilation significantly improved the modelled SWE and PET despite the poor performance in some regions of the study area (see Figure 2). The model failure in some regions was partly due to the datasets, e.g., the passive microwave snow product in mountainous terrain and coastal areas or when snow is deep. The MODIS evapotranspiration is also known to be consistently biased [10] although it is also possible that the model was unable to properly represent processes in those regions.

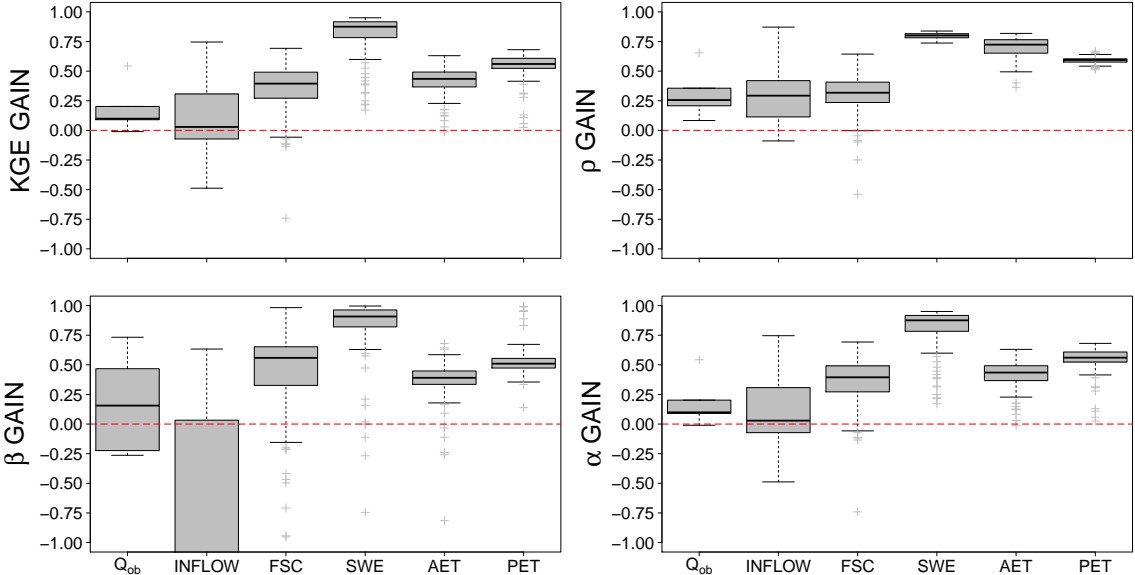

**Figure 4.** The gain in the KGE and improvements in its components with each of the six datasets assimilated alone. The KGE was computed with the model outputs of the assimilated variables themselves. The red lines represent the open-loop simulation i.e., with zero gain.

### 4.4. Assimilation of Combined Datasets

Here, we assimilated different multi-variable combinations and assessed the gain in performance for the target variables. Figure 5 shows the gain in the KGE for the the discharge with the assimilations along the horizontal axis of the figure. The combinations are given in Table A1. The combinations that improved the discharge predictions are presented in Table 2. The INFLOW was present in all combinations apart from when $Q_{ob}$ was assimilated alone. The ET products appeared in five out of the 63 assimilations while the SWE appeared in only three combinations. The SWE data in the mountainous and coastal areas of the basin were excluded from the assimilation due to poor quality and this lack of data in those regions could have caused the poor SWE performance. The assimilation routine solely relied on the value of the prescribed characteristic length, which might not have allowed adequate propagation of snow information to those areas.

**Table 2.** The assimilations (numbers in the first row, products are the crosses down the columns) that resulted into gains for discharge predictions.

| Assim. | 12 | 22 | 7 | 39 | 36 | 14 | 2 | 42 | 53 | 43 | 26 | 25 | 19 | 33 | 23 | 59 | 1 |
|---|---|---|---|---|---|---|---|---|---|---|---|---|---|---|---|---|---|
| GAIN($\times10^{-2}$) | 25 | 11 | 10 | 6 | 5.2 | 5 | 3 | 2.9 | 2.5 | 2 | 1.9 | 1.6 | 1.5 | 0.9 | 0.8 | 0.2 | 0.2 |
| $Q_{ob}$ | + | + |  | + |  |  |  | + | + | + |  | + |  | + |  | + | + |
| INFLOW | + | + | + | + | + | + | + | + | + | + | + | + | + | + | + | + | + |
| FSC |  | + | + |  | + |  |  | + | + | + | + |  |  |  | + |  | + |
| SWE |  |  |  |  |  |  |  | + |  |  |  | + |  |  | + |  |  |
| AET |  |  |  |  |  | + |  |  |  | + | + |  |  | + |  | + |  |
| PET |  |  |  | + | + |  |  |  | + |  |  |  | + |  |  | + |  |

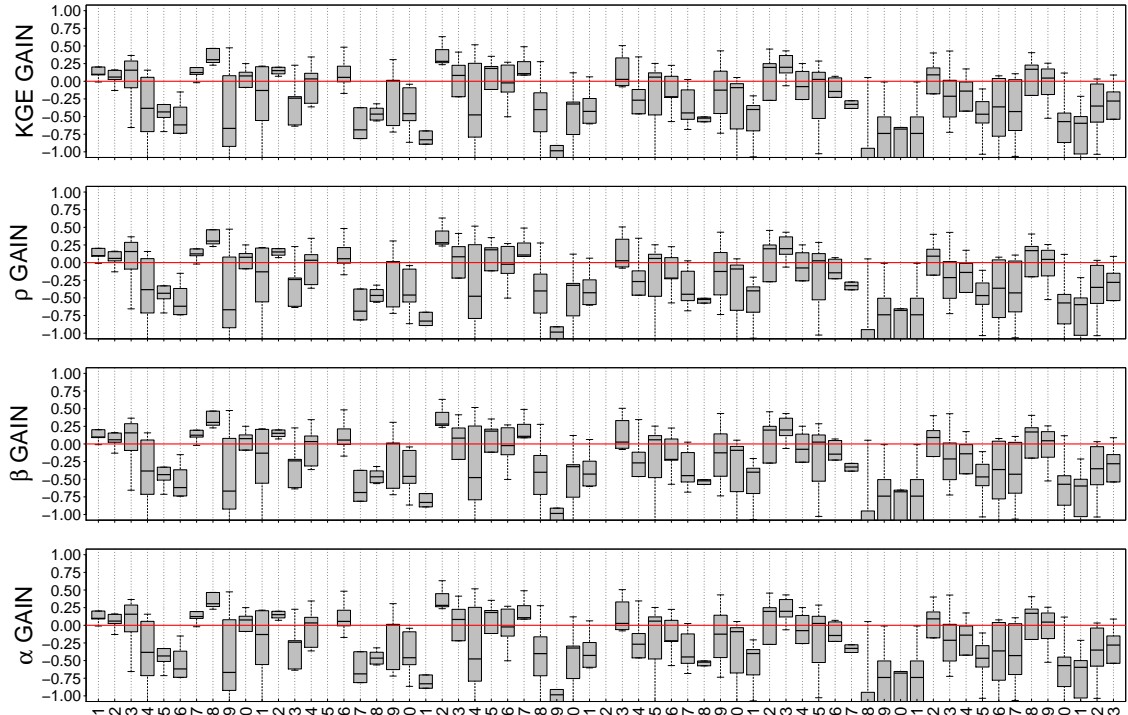

**Figure 5.** The gain in the KGE and its components computed with the stream discharge from the different assimilations. The numbers on the x-axis are the assimilations enumerates in Table A1 and the red lines are the open-loop simulations.

Assimilation of PET improved the KGE in five (19, 36, 39, 53 and 59) out of the 31 assimilations (6, 15, 18, 19–21, 29–32, 36–41, 46–51, 53–57 and 59–63) it was present. It however improved the $Q_{ob}$ bias and variability. The AET produced KGE gains in five out of the 32 assimilations it was present. The ET products never produced gains when assimilated individually, or in combinations involving either one or both together with SWE. This underpinned the lesser role played by these satellite products in assimilations aimed at improving discharge predictions. Therefore, the fitness for purpose of satellite ET in hydrology could lie elsewhere, e.g., in reducing the parameter search space [16]. The passive microwave SWE cannot be used in its current state without addressing the limitations in complex terrain and coastal regions.

Figure 6 shows the improvements in KGE components for the reservoir inflow predicted from different combinations. The combinations resulting into KGE gain are tabulated in Table 3.

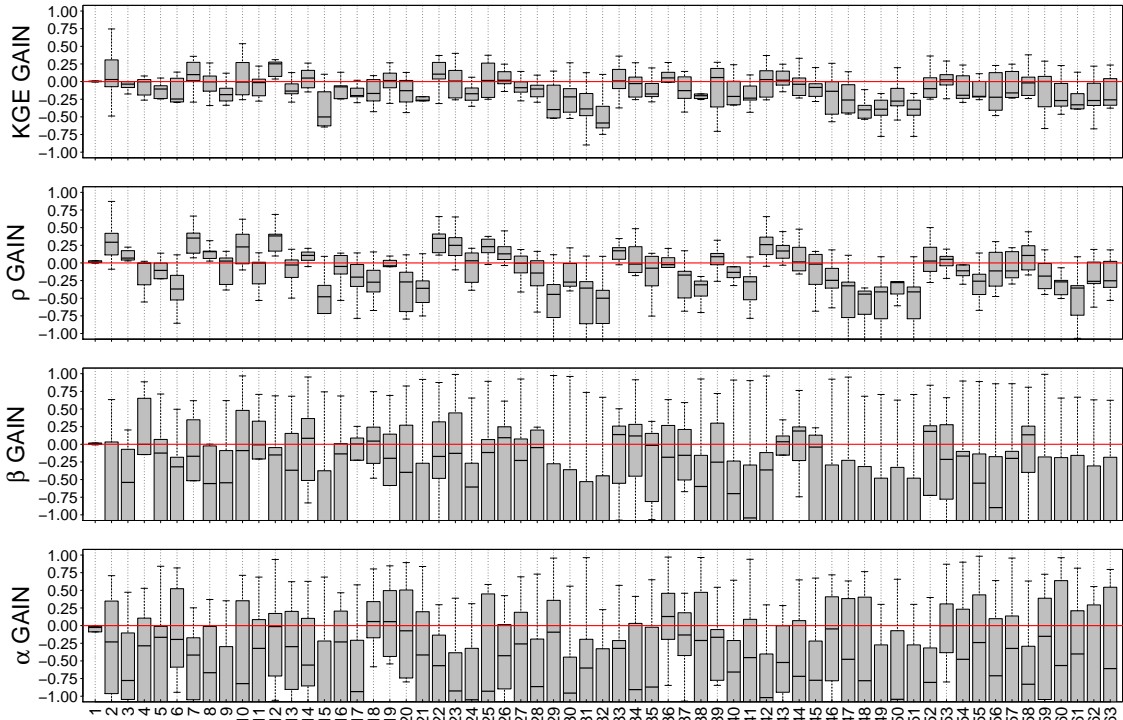

**Figure 6.** The gain in the KGE and its components computed with the reservoir inflow from the different assimilations. The numbers on the x-axis are the assimilations enumerates in Table A1 and the red lines are the open-loop simulations.

**Table 3.** The product combinations (assimilations numbers from Table A1 in the first row, products are the crosses down the columns) that resulted into gains for reservoir inflow predictions.

| Assim. | 8 | 22 | 43 | 42 | 25 | 58 | 3 | 12 | 7 | 27 | 1 | 52 | 23 | 10 | 2 | 35 | 16 | 59 | 14 | 33 | 45 | 19 |
|---|---|---|---|---|---|---|---|---|---|---|---|---|---|---|---|---|---|---|---|---|---|---|
| GAIN($\times 10^{-2}$) | 30 | 28 | 20 | 19 | 18 | 17 | 16 | 15 | 12 | 11 | 10 | 9 | 8 | 7 | 6 | 6 | 5 | 4 | 3 | 3 | 2.6 | 0.46 |
| $Q_{ob}$ | + | + | + | + | + | | + | | + | + | + | | | | | + | + | + | | + | + | |
| INFLOW | | + | + | + | + | | + | + | | | | + | + | + | | | + | + | + | | | + |
| FSC | + | + | + | + | + | + | | + | + | | | | + | | | | | + | | | + | |
| SWE | | | + | | + | | | | | | | + | + | + | | + | | | | | + | |
| AET | | | + | | | | | | | + | | + | | | + | + | + | + | + | | | |
| PET | | | | | | | | | | | | | | | | | + | | | | | + |

The assimilation of PET resulted in KGE gains in just two combinations that did not include SWE. AET produced gains when combined with SWE in four assimilations, which again showed that those datasets should not be combined. The combinations differ from those for discharge, therefore the optimal combinations depend on the selected performance metric as well as the target variable.

We designated the 13 combinations tabulated in Table 4 that resulted in KGE gains for both discharge and reservoir inflow predictions as *informative*. More combinations would have qualified if one target variable had been considered. INFLOW was present in all but the one informative assimilation involving $Q_{ob}$ alone. PET was present in two, the fewest informative assimilations for a dataset.



**Table 4.** The identified informative combinations of products (crosses) for assimilation.

| Assim. | $Q_{ob}$ | INFLOW | FSC | SWE | AET | PET |
|--------|----------|--------|-----|-----|-----|-----|
| 1 | + | | | | | |
| 2 | | + | | | | |
| 7 | | + | + | | | |
| 12 | + | + | | | | |
| 14 | | + | | | + | |
| 19 | | + | | | | + |
| 22 | + | + | + | | | |
| 23 | | + | + | + | | |
| 25 | + | + | | + | | |
| 33 | + | + | | | + | |
| 42 | + | + | + | + | | |
| 43 | + | + | + | | + | |
| 59 | + | + | + | | + | + |

An assessment of the individual product's performance in assimilations is summarised in Table 5. The integers are the number of times assimilations containing a particular product either alone or combined with others resulted in a KGE gain. The products generally performed better in discharge predictions. The overall ranking from the best was: INFLOW, $Q_{ob}$, FSC, AET, SWE and PET. In situ measurements that aggregate information from all the upstream contributing areas outperformed satellite data. The FSC was the top performing satellite product due to the detailed metadata included with the dataset. We used the metadata quality flags to set acceptability thresholds, which ensured the accuracy of the assimilated data. The least performing PET outperformed the SWE and INFLOW in improving the discharge bias. This implies that the PET dataset might be useful for applications other than data assimilation, e.g., water balance calibration.

**Table 5.** The number of times product assimilations produced improvements in the median KGE, $\rho$, $\beta$ and $\alpha$ for the predicted discharge and reservoir inflow (subscripts Q and INFLOW, respectively).

| | $KGE_Q$ | $KGE_{INFLOW}$ | $\rho_Q$ | $\rho_{INFLOW}$ | $\beta_Q$ | $\beta_{INFLOW}$ | $\alpha_Q$ | $\alpha_{INFLOW}$ |
|-----------|---------|----------------|----------|-----------------|-----------|------------------|------------|-------------------|
| $Q_{ob}$ | 15 | 10 | 18 | 13 | 17 | 5 | 9 | 0 |
| Inflow | 16 | 16 | 17 | 15 | 8 | 5 | 14 | 2 |
| FSC | 11 | 9 | 15 | 13 | 14 | 5 | 14 | 2 |
| SWE | 8 | 3 | 12 | 9 | 11 | 5 | 5 | 0 |
| AET | 10 | 5 | 10 | 7 | 13 | 9 | 11 | 0 |
| PET | 2 | 5 | 2 | 2 | 12 | 1 | 6 | 3 |

The predictions of the monthly stream discharge for the ensemble of 13 informative assimilation combinations during the year 2001 are shown in Figure 7. The assimilations were better than the open loop and had reasonable predictions at all gauged stations. The July discharge was underestimated in all assimilations at the headwater stations 8313880 and 8301032 where the SWE was excluded from the assimilations. This might mean an underestimation of the snow melt.

Finally, we investigated the KGE gain at a monthly scale for the target variables as shown in Figure 8. With the exception of stream discharge, all informative assimilations improved the reservoir inflows during the snow-melt period April to July. This is especially important for the power generation companies who strive to balance the volume of water to run through the plants for continuous energy production against the volume to be stored in reservoirs for winter production. Assimilation of the informative combinations initialised during the snow-melt period would guarantee improved forecasts of the reservoir inflows, hence better decisions regarding apportioning of reservoir water storage. The improvements in the correlation, variability and bias for the ten assimilations are included in Figures A1, A2 and A3 in Appendix A.

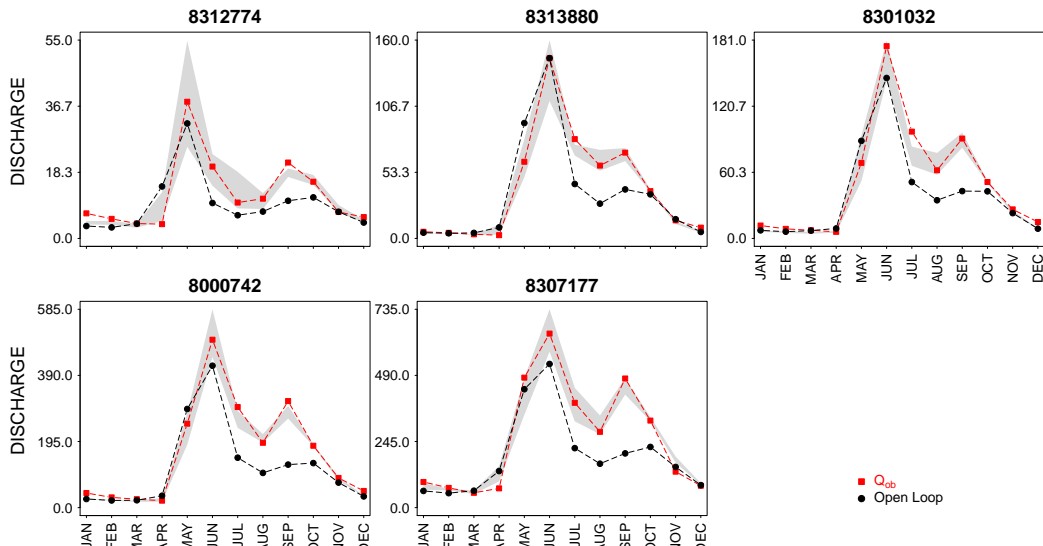

**Figure 7.** The simulated average monthly discharges during 2001 at the calibration stations (IDs are the sub-figure titles) for the informative assimilations ensemble (filled polygon) compared to the observations (red) and open-loop simulation (black). The location of the stations in the study area is shown in Figure 1.

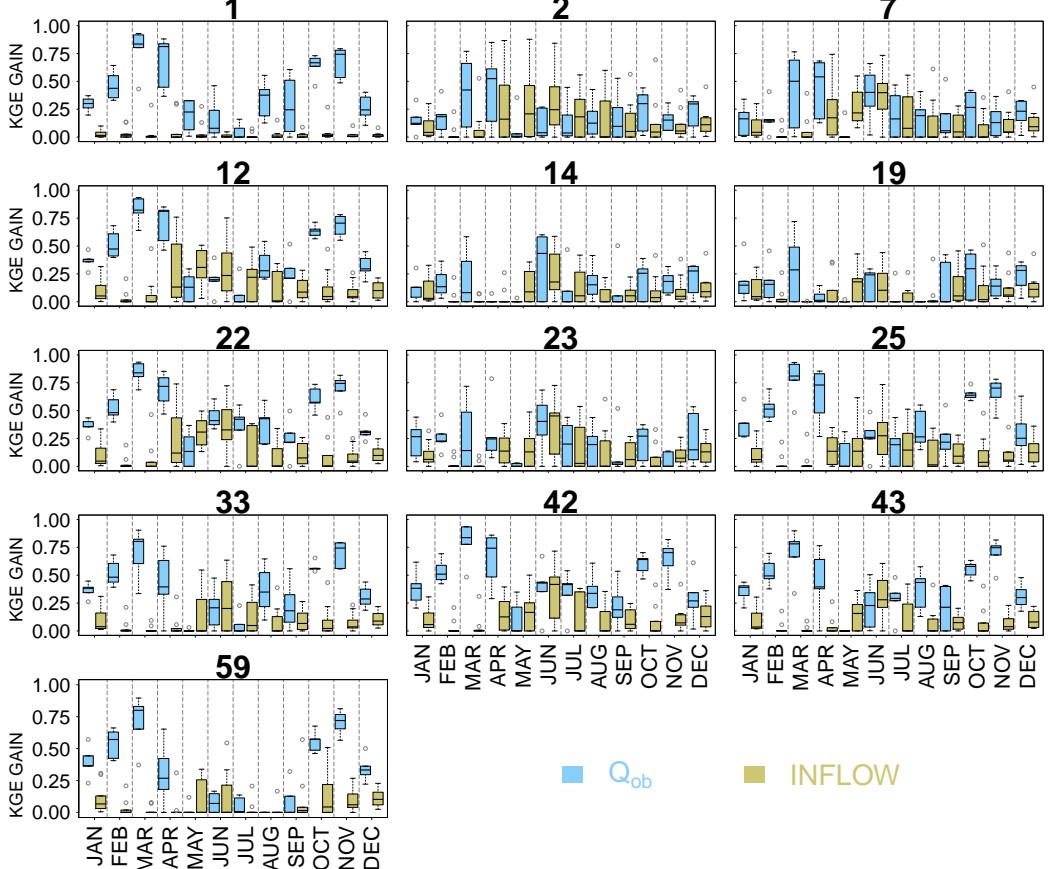

**Figure 8.** The monthly gains in the KGE computed for stream discharge (blue) and reservoir inflow (green) for informative assimilations (titles of the individual sub-figures are the row numbers taken from Table A1).

## 5. Discussion

Satellite data fill the spatial and temporal gaps in ground-based observations and also provide models with initial and boundary conditions. Data assimilation is an elegant way of incorporating the satellite observations to produce consistent model outputs at large and continuous scales in space and time. Depending on the satellite footprint and the length of the observations, this presents a very wide range of possibilities for example in regions with sparse to non existent observation networks. Data assimilation improves the internal model states and fluxes, whose time-shots provide initialisation points for either reconstructing known historical events (hindcast) or forecasting future events, e.g., in Numerical Weather Predictions (NWP).

Our current experiments have shown an additional value of data assimilation; namely, the prediction of summer-time snow melt in a Swedish catchment. This is of practical importance since very large sums of money can potentially be lost if the power companies generate power during the summer when the prices are low, or when they have to operate the spillways as a result of erroneous reservoir inflow forecasts.

One then needs to translate the gains in the reservoir inflow predictions into economic (monetary) gains. For this an economic model that transforms the flow through the turbines into revenue per unit of energy produced is required. The derivation of such a model is a subject of ongoing research.

## 6. Conclusions

Here, we assimilated six (two ground-based and four satellite) products at different spatial scales into a hydrological model aiming to (1) appraise the performance of the assimilation routine and (2) determine the optimum ensemble size of the Kalman filter. We also assimilated a further 57 combinations of those six products to determine the product combinations that improved the predictions of both stream discharge and reservoir inflow. The investigations led us to the following:

1. Ensembles with 25 members were sufficient for all but $Q_{ob}$ and INFLOW assimilations which required 100 members. The assimilation exercise led to significant improvements in the model's prediction of both the PET and SWE. This somewhat addressed the problems endemic to the two datasets i.e., very high PET and erroneous SWE in the mountains and along the coast.
2. The assimilation of the MODIS PET product did not improve the stream discharge and reservoir inflow predictions, which points to the limited value of the product in data assimilation. AET performed better than PET but still worse than the satellite snow products. It would be interesting to assimilate different ET products e.g., the GLEAM (Available online: https://www.gleam.eu/ (accessed on 8 May 2012).) and Land-SAF (Available online: https://landsaf.ipma.pt/en/products/evapotranspiration-energy-flxs/ (accessed on 8 May 2012).) to ascertain if the poor performance of the assimilations was limited to the MODIS products.
3. In situ observations' assimilations outperformed the satellite products apart from the FSC that was sometimes better than the discharge. The main difference between the FSC and other satellite datasets was the extensive quality flags that enabled its customisation to fit local conditions. This highlights the very critical sensitivity of data assimilation to the quality of assimilated data.
4. From the 63 assimilation experiments, we obtained 13 combinations that had median KGE's above the open-loop for both the stream discharge and the reservoir inflow. Apart from the stream discharge, those assimilations improved the predicted reservoir inflow during the snow melt months. This is important for hydropower companies which need reliable forecasts of the snow-melt runoff to decide whether to use water for continuous energy production or store it for winter energy production.

**Author Contributions:** Conceptualization, D.G. and I.P.; Methodology, J.L.M., I.P., D.G., L.C., R.P.; Software, J.L.M., D.G.; Validation, J.L.M., D.G., R.P.; Formal Analysis, J.L.M., R.P.; Investigation, J.L.M., R.P., D.G.; Writing-Original Draft Preparation, J.M.; Writing-Review & Editing, J.L.M., L.C., R.P., D.G., I.P.; Visualization, J.L.M.; Supervision, I.P., D.G.; Project Administration, I.P.; Funding Acquisition, D.G., I.P. All authors have read and agreed to the published version of the manuscript.

**Funding:** This study was partially funded by the EU Horizon 2020 project IMPREX (Improving predictions and management of hydrological extremes) under grant agreement No. 641811. We also received partial funding from the EU Horizon 2020 project SPACE-O (Space Assisted Water Quality Forecasting Platform for Optimized Decision Making in Water Supply Services) under grant agreement No. 730005. This study was also partially funded by the EU Horizon 2020 project PrimeWater (Delivering advanced predictive tools from medium to seasonal range for water dependent industries exploiting the cross-cutting potential of EO and hydro-ecological modelling) under grant agreement No. 870497. Finally, this study was partially funded by the EU Horizon 2020 project E-SHAPE (EuroGEOSS Showcases: Applications Powered by Europe) under grand agreement No. 820852.

**Acknowledgments:** We performed the investigations at the SMHI Hydrology Research unit, where work benefits from joint efforts in developing models and concepts by the whole team. The HYPE model code is available from the HYPEweb portal (Available online: http://hypeweb.smhi.se/model-water/ (accessed on 8 May 2012).).

**Conflicts of Interest:** The authors of this work declare no conflict of interest. Our funders had no role in the choice of research project; design of the study; in the collection, analyses or interpretation of data; in the writing of the manuscript; or in the decision to publish the results.

## Appendix A

*Appendix A.1. The Ensemble Kalman Filter for Data Assimilation*

Let $\mathbf{X}^b$ be the $N_{state} \times N_{ens}$ matrix of background model states, where $N_{state}$ is the number of state variables and $N_{ens}$ is the number of ensemble members.

$$\mathbf{X}^b = (\mathbf{x}_1^b, ..., \mathbf{x}_{N_{ens}}^b),$$ (A1)

where $\mathbf{x}_1^b, ..., \mathbf{x}_{N_{ens}}^b$ are the background vectors of all model states for each of the $N_{ens}$ ensemble members before the update. The ensemble mean is defined as

$$\bar{\mathbf{x}}^b = \frac{1}{N_{ens}} \sum_{i=1}^{N_{ens}} \mathbf{x}_i^b.$$ (A2)

The ensemble anomaly matrix is defined as $\mathbf{Z}^b = (\mathbf{z}_1^b, ..., \mathbf{z}_{N_{ens}}^b)$, where the ensemble anomaly for the $i^{th}$ member is $\mathbf{z}_i^b = \mathbf{x}_i^b - \bar{\mathbf{x}}^b$. The estimate of the model error covariance is computed from the ensemble anomalies as

$$\mathbf{P}^b = \frac{1}{N_{ens} - 1} \mathbf{Z}^b \mathbf{Z}^{b^T},$$ (A3)

where $(\cdot)^T$ denotes the matrix transpose. The model states are then updated by following the equations in Magnusson et al. [12], Clark et al. [31], Turner et al. [56]

$$\mathbf{x}_i^a = \mathbf{x}_i^b + \mathbf{K}(\mathbf{y}_i - \mathbf{H}\mathbf{x}_i^b).$$ (A4)

where $\mathbf{x}_i^a$ is the analysis of model states after the update, $\mathbf{y}_i$ is the vector of observations with length $N_{obs}$, $\mathbf{H}$ is the $N_{obs} \times N_{state}$ observation operator that maps the model states onto the observation space and $\mathbf{K}$ is the Kalman gain. It should be noted that each ensemble member is updated separately in Equation (A4). The Kalman gain $\mathbf{K}$ is computed from

$$\mathbf{K} = \mathbf{P}^b \mathbf{H}^T \left[ \mathbf{H}\mathbf{P}^b\mathbf{H}^T + \mathbf{R} \right]^{-1},$$ (A5)

where $\mathbf{R}$ is the $N_{obs} \times N_{obs}$ observation error covariance matrix. The gain is composed of covariance matrices as in Magnusson et al. [12]: $\mathbf{C}_{xy} = \mathbf{P}^b\mathbf{H}^T$, the covariance matrix between the errors of states and predicted measurements and $\mathbf{C}_{yy} = \mathbf{H}\mathbf{P}^b\mathbf{H}^T$, the covariance matrix between the errors of the predicted measurements themselves. The optimisation of the Kalman gain is achieved by minimising the analysis error [24,57], which is written as

$$\boldsymbol{\epsilon}^a = \boldsymbol{\epsilon}^b + \mathbf{K}(\boldsymbol{\epsilon}^o - \mathbf{H}\boldsymbol{\epsilon}^b),$$ (A6)

where the superscripts *a* , *b* , *o* denote the analysis, background and observation, respectively. The errors are represented by their covariance matrices, which are composed of covariance errors (variance errors along the diagonals). They are written as $\mathbf{A} = E[\epsilon^a \epsilon^{a^T}]$; $\mathbf{B} = E[\epsilon^b \epsilon^{b^T}]$; and $\mathbf{R} = E[\epsilon^o \epsilon^{o^T}]$ for the analysis, background and observations, respectively and $E[\cdot]$ represents the expected value. The gain matrix in (A5) can potentially cause numerical problems if the inverted term becomes very small. We overcame this by fixing the minimum allowable error variance.

Aalstad et al. [4] uses analytical Gaussian anamorphosis to update physically bounded variables. Moreover, the updated state should be physically consistent to be fit for the purpose of further propagation into the model. Bertino et al. [34] found that the model could give non-physical results if invalid physical states were provided for the restart. They propose the *lognormal* and *logit-normal* transformations for semi-bounded and bounded variables, respectively and define the logit-normal transformation for a variable *x* bounded between *a* and *b* as

$$\tilde{x} = \text{logit}_{(a,b)}(x) = \ln\left(\frac{x-a}{b-a}\right) - \ln\left(1 - \frac{x-a}{b-a}\right). \tag{A7}$$

Please see Bertino et al. [34] for a detailed discussion of the transformations and the underlying assumptions.

*Appendix A.2. Supplementary Material*

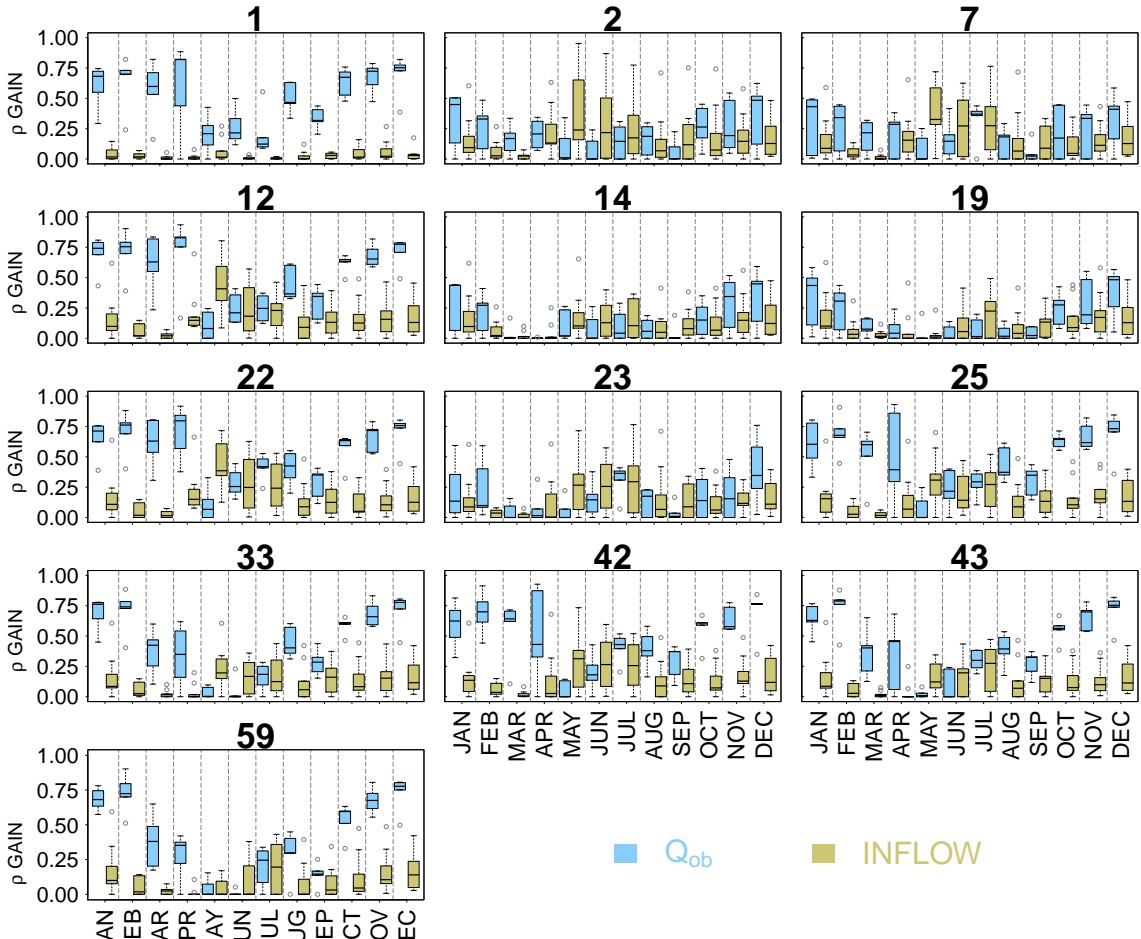

**Figure A1.** The monthly improvements in the correlation computed for stream discharge (blue) and reservoir inflow (green) for informative assimilations (titles of the individual sub-figures are the row numbers taken from Table A1).

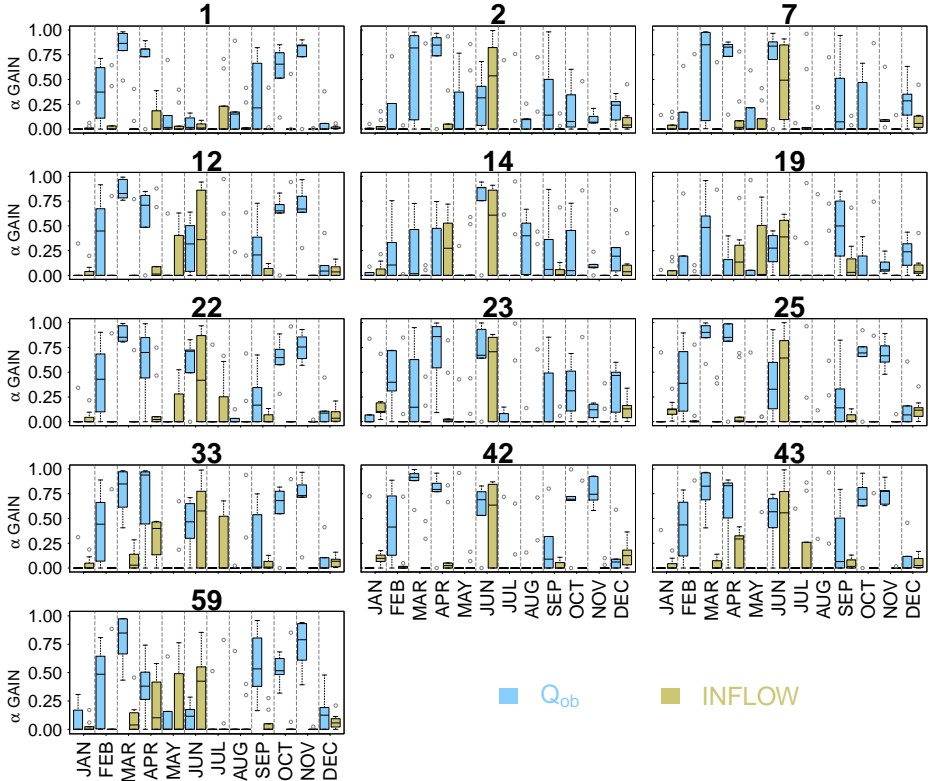

**Figure A2.** The monthly improvements in the variability computed for stream discharge (blue) and reservoir inflow (green) for informative assimilations (titles of the individual sub-figures are the row numbers taken from Table A1).

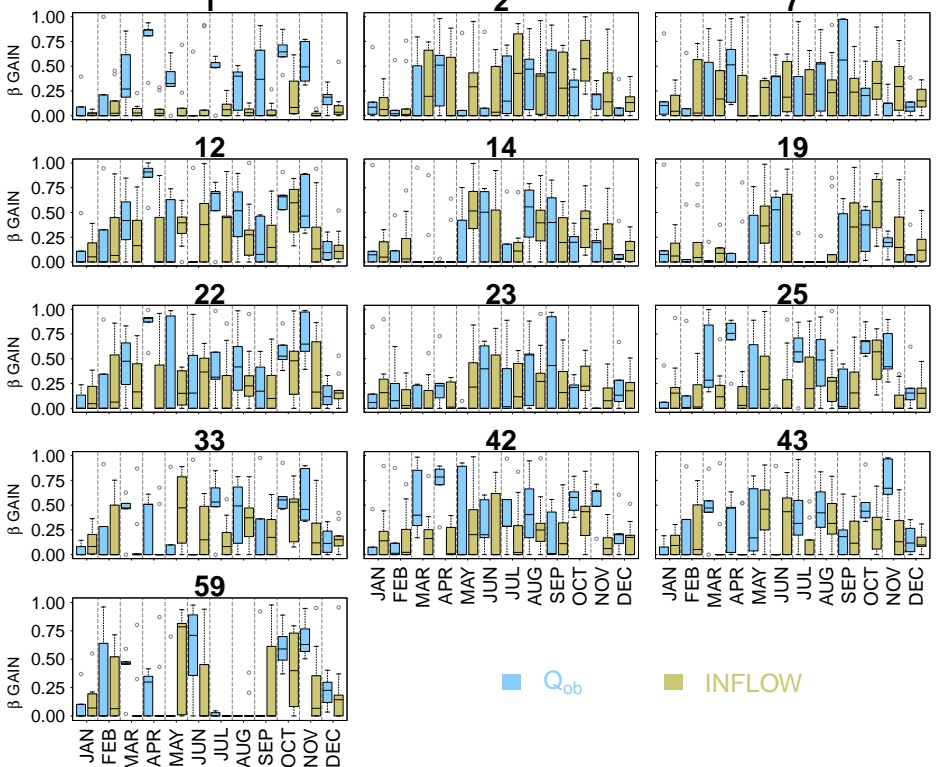

**Figure A3.** The monthly improvements in the bias computed for stream discharge (blue) and reservoir inflow (green) for informative assimilations (titles of the individual sub-figures are the row numbers taken from Table A1).

**Table A1.** The assimilations performed in the experiment. The crosses show the product combinations and the highlighted numbers are the informative ones with median KGE gains for both discharge ($\Omega_Q$) and reservoir inflow ($\Omega_{\text{INFLOW}}$). The colour codes are enumerated in the legend at the bottom of the table.

| Assim. | $Q_{ob}$ | INFLOW | FSC | SWE | AET | PET | $\Omega_Q$ | $\Omega_{\text{INFLOW}}$ |
|---|---|---|---|---|---|---|---|---|
| 1 | + | | | | | | blue | blue |
| 2 | | + | | | | | blue | blue |
| 3 | | | + | | | | red | cyan |
| 4 | | | | + | | | red | red |
| 5 | | | | | + | | red | red |
| 6 | | | | | | + | red | red |
| 7 | | + | + | | | | blue | cyan |
| 8 | + | | + | | | | red | green |
| 9 | | | + | + | | | red | red |
| 10 | | + | | + | | | red | blue |
| 11 | + | | | + | | | red | red |
| 12 | + | + | | | | | yellow | cyan |
| 13 | | | + | | + | | red | red |
| 14 | | + | | | + | | blue | blue |
| 15 | | | | | + | + | red | red |
| 16 | + | | | | + | | red | blue |
| 17 | | | + | | + | | red | red |
| 18 | | | + | | | + | red | red |
| 19 | | + | | | | + | blue | blue |
| 20 | + | | | | | + | red | red |
| 21 | | | | + | | + | red | red |
| 22 | + | + | + | | | | cyan | green |
| 23 | | + | + | + | | | blue | blue |
| 24 | + | | + | + | | | red | red |
| 25 | + | + | | + | | | blue | cyan |
| 26 | | + | + | | + | | blue | red |
| 27 | + | | + | | + | | red | blue |
| 28 | | | + | + | + | | red | red |
| 29 | | | + | | + | + | red | red |
| 30 | | + | | | + | + | red | red |
| 31 | + | | | | + | + | red | red |
| 32 | | | | + | + | + | red | red |
| 33 | + | + | | | + | | blue | blue |
| 34 | | + | | + | + | | red | red |
| 35 | + | | | + | + | | red | blue |
| 36 | | + | + | | | + | blue | red |
| 37 | + | | + | | | + | blue | red |
| 38 | | | + | + | | + | red | red |
| 39 | + | + | | | | + | blue | red |
| 40 | | + | | + | | + | red | red |
| 41 | + | | | + | | + | red | red |
| 42 | + | + | + | + | | | blue | cyan |
| 43 | + | + | + | | + | | blue | cyan |
| 44 | | + | + | + | + | | red | red |
| 45 | + | | + | + | + | | red | blue |
| 46 | | + | + | | + | + | red | red |
| 47 | + | | + | | + | + | red | red |
| 48 | | | + | + | + | + | red | red |
| 49 | + | + | | | + | + | red | red |
| 50 | | + | | + | + | + | red | red |
| 51 | + | | | + | + | + | red | red |
| 52 | + | + | | + | + | | red | blue |
| 53 | + | + | + | | | + | blue | red |
| 54 | | + | + | + | | + | red | red |
| 55 | + | | + | + | | + | red | red |
| 56 | + | + | | + | | + | red | red |
| 57 | + | + | + | + | | + | red | red |
| 58 | + | + | + | + | + | | red | cyan |
| 59 | + | + | + | | + | + | blue | blue |
| 60 | | + | + | + | + | + | red | red |
| 61 | + | | + | + | + | + | red | red |
| 62 | + | + | | + | + | + | red | blue |
| 63 | + | + | + | + | + | + | red | red |

● $\Omega < 0.00$; ● $0.00 \leq \Omega < 0.10$; ● $0.10 \leq \Omega < 0.20$; ● $0.20 \leq \Omega < 0.25$; ● $0.25 \leq \Omega < 0.30$; ● $\Omega \geq 0.30$

*Appendix A.3. Parameters Controlling the Ensemble Kalman Filter*

A brief description of the parameters controlling the ensemble Kalman filter performance is presented below.

| | |
|---|---|
| **EnsTyp** | The type of the ensemble bounds: 1 = unrestricted, 2 = semi-restricted(minimum), 3 = semi-restricted(maximum), 4 = constrained (max and min). |
| $\sigma$ | The constant standard deviation used for EnsTyp = 1, also used as minimum allowed standard deviation for EnsTyp = 2–4 |
| **SMeta** | Relative standard deviation for EnsTyp = 2–3 |
| **RMeta** | Relative standard deviation for EnsTyp = 4 |
| **LScale** | Length scale in the generation of spatially correlated fields. 0 = uncorrelated fields. |
| **Cell** | Cell size (x and y directions) in the 2D grid used for the spatially correlated random fields (interpolated to the model coordinates). 0 = uncorrelated fields. |
| **CTyp** | Correlation function: 0 none 1 Gaussian, 2 Compact 5th degree polynomial, 3 Power law |
| **CoordID** | spatial domain of observation 1 = subbasin, 2 = upstream area (discharge), 3 = aquifer, 4 = outregions (inflow) |
| **Trans** | Transformation of the observations 0 = none, 1 = log, 3 = logit |
| $\epsilon$ | Minimum value used to avoid 0 in log or logit transformations. |

**Table A2.** The parameters controlling the ensemble Kalman filter in data assimilation

| Var. | EnsTyp | $\sigma$ | SMeta | RMeta | LScale | Cell | CTyp | CoordID | Trans | $\epsilon$ |
|---|---|---|---|---|---|---|---|---|---|---|
| $Q_{ob}$ | 2 | 0.1 | 0.3 | 0.15 | 0 | 0 | 0 | 2 | 1 | $1 \times 10^{-6}$ |
| INFLOW | 1 | 15 | 0.3 | 0.15 | 0 | 0 | 0 | 4 | 0 | $1 \times 10^{-6}$ |
| FSC | 4 | 0.01 | 0.15 | 0.15 | 0 | 0 | 0 | 1 | 3 | $1 \times 10^{-6}$ |
| SWE | 2 | 0.01 | 0.3 | 0.15 | 0 | 0 | 0 | 1 | 1 | $1 \times 10^{-6}$ |
| AET | 2 | 0.01 | 0.15 | 0.15 | 0 | 0 | 0 | 1 | 1 | $1 \times 10^{-6}$ |
| PET | 2 | 0.01 | 0.15 | 0.15 | 0 | 0 | 0 | 1 | 1 | $1 \times 10^{-6}$ |
| Temp | 1 | 2 | 0.3 | 0.15 | $5 \times 10^4$ | $1 \times 10^4$ | 1 | - | - | - |
| Precip | 2 | 0.01 | 0.3 | 0.15 | $5 \times 10^4$ | $1 \times 10^4$ | 1 | - | - | - |

The ensemble size, (varied in this study) and the horizontal and vertical length scales also play a role. Length scales are the distances at which a reduction of approximately 90% in the covariance occurs, they were set to 50 and 1km in the horizontal and vertical directions, respectively.

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
