# Peer review of "Impact of Satellite and In Situ Data Assimilation on Hydrological Predictions"

_remotesensing, doi:10.3390/rs12050811_

Round 1

Reviewer 2 Report

This paper has the potential to be an excellent reference for hydrological data assimilation therefore it is important that the material in the results and discussion be better organized.  Many discussion points are interspersed among the reporting of results.  This leads to the points being brought forth as side note observations and leaves them inadequately discussed.  This is a pity because there are many insightful reflections made throughout the result section that would ordinarily be addressed in the discussion section, which in the current version is very sparse and lacking substance.

Overall the paper is an impressive data assimilation effort.  The work could be of high value to researchers seeking tools to better understand dynamic processes of reservoir based hydrologic systems. As such, it is vital that the introduction clearly define the scope of the "tool" i.e. data assimilation. Data assimilation has a rich foundation but it is a very specific tool. As my comments to L52 and L54 suggests, a better description of the "tool" will help provide context, especially to those not intimately familiar with data assimilation.

L52 - please define "data assimilation." Your description of the result you focus on (e.g. "choose the final estimate such that the uncertainty of the final state is minimized"  needs more context.

L54 - furnished is an odd word choice. Ordinarily I would suggest changing "furnished" to achieved, accomplished, conducted, or performed; however, I think your intended meaning aligns with the fact that data assimilation "furnishes" an estimate of how far a model's output deviates from the "true" state thereby providing a means for correcting the deviation. However, without a better description of "what" data assimilation is or does the use of furnished could confuse readers not deeply familiar with data assimilation.

L66-68 - The statement is not substantiated with citations. It is highly debatable and adds little to the explanation about the research being undertaken or the research results.

L74- be specific and accurate. you aren't improving the variables you are seeking to minimize the difference between model and measurement of river discharge and reservoir inflow.

L82 -odd wording. Consider a more standard phrasing such as " To the best of our knowledge this work is the first to conduct data assimilation of multiple data products...

L134- odd wording (so-called)- consider more standard phrasing such as " Local inflow is defined as the sum of runoff...

L136-137 awkward, hard to follow the precise meaning of this sentence.

L197-198 typos. lack of object.. correlation was (what?). consider changing to "Variability and AET correlation were also...

Figure 1. The map in this figure is sorely inadequate. It lacks a distance measure, lat/long or other contextual guide such as a map inset. The boundaries of the colored "polygons" aren't coincident with the "sub-basin polygons."
Figure 2. Why are the maps for Inflow (second row) so different from the other rows? It makes it difficult to examine cross variable values for specific Umeälven sub-basins.

All graphs in Figures 3 - 7 need the y-axis labeled
